# Monogenic Disorders of ROS Production and the Primary Anti-Oxidative Defense

**DOI:** 10.3390/biom14020206

**Published:** 2024-02-09

**Authors:** Nana-Maria Grüning, Markus Ralser

**Affiliations:** 1Department of Biochemistry, Charité Universitätsmedizin Berlin, 10117 Berlin, Germany; markus.ralser@charite.de; 2The Wellcome Centre for Human Genetics, Nuffield Department of Medicine, University of Oxford, Oxford OX3 7BN, UK; 3Max Planck Institute for Molecular Genetics, 14195 Berlin, Germany

**Keywords:** oxidative stress, reactive oxygen species (ROS), cellular redox balance, monogenic disorder, inherited disease

## Abstract

Oxidative stress, characterized by an imbalance between the production of reactive oxygen species (ROS) and the cellular anti-oxidant defense mechanisms, plays a critical role in the pathogenesis of various human diseases. Redox metabolism, comprising a network of enzymes and genes, serves as a crucial regulator of ROS levels and maintains cellular homeostasis. This review provides an overview of the most important human genes encoding for proteins involved in ROS generation, ROS detoxification, and production of reduced nicotinamide adenine dinucleotide phosphate (NADPH), and the genetic disorders that lead to dysregulation of these vital processes. Insights gained from studies on inherited monogenic metabolic diseases provide valuable basic understanding of redox metabolism and signaling, and they also help to unravel the underlying pathomechanisms that contribute to prevalent chronic disorders like cardiovascular disease, neurodegeneration, and cancer.

## 1. Introduction

Reactive oxygen species (ROS, [1]) are formed by incomplete reduction of oxygen, and comprise a variety of highly reactive, oxygen-containing compounds, with the superoxide anion radical (O_2_**^·^**^−^), hydrogen peroxide (H_2_O_2_) and the hydroxyl radical (OH**^·^**) being the most prominent types in aerobic organisms [2]. ROS can be subdivided into free radicals and nonradicals. The former carry an unpaired electron (e.g., O_2_**^·^**^−^, OH**^·^**) and can combine to form nonradicals (e.g., H_2_O_2_) [3]. Cells can experience ROS exposure either through their production by endogenous systems or through external perturbations. At physiological concentrations, ROS serve important functions.For example, H_2_O_2_ acts as a second messenger to communicate pro-inflammatory and growth-stimulating signals [4,5,6]. Furthermore, ROS are also employed in the defense against pathogens when immune cells produce high levels of O_2_**^·^**^−^ or H_2_O_2_ to kill bacteria or fungi [7].

However, in order to stabilize themselves, ROS take up electrons and oxidize, and thereby unspecifically damage biomolecules such as nucleic acids, metabolites, lipids, and proteins. In order to keep oxidative damage low, cells have evolved anti-oxidative defense systems. When the level of ROS overflows the capacity of these defense mechanisms, oxidative stress occurs, with the redox balance of the cell, or of its different compartments, shifting away from the physiologically ideal state. If this deviation is strong, disruption of cellular processes may occur [8]. Hence, oxidative stress has implications in various human diseases such as cancer, stroke, or late-onset neurodegenerative disorders caused by environmental stresses or congenital dysfunction of the proteins involved in redox balance [9]. 

When endogenous ROS production or the anti-oxidative machinery are inherently impaired, the cell has limited capacities to tackle oxidative stress from the start. In this review, we summarize the main monogenic disorders that are caused by germline pathogenic variants in genes that are part of endogenous ROS production and the primary anti-oxidative defense machinery. Our emphasis is on the initial encounters of cells with ROS, encompassing ROS production, scavenging, and the recycling of ROS scavengers. While repair mechanisms for damaged biomolecules associated with prolonged adaptation to oxidative stress are vital components, their abundance demands dedicated exploration beyond the scope of this review. 

This overview of monogenic disorders related to ROS production and the primary anti-oxidative defense encompasses a spectrum ranging from prevalent enzymopathies to some of the rarest inherited disorders. Putting the spotlight onto these rare enzymopathies describes the intricacies of redox metabolism and might inspire more research attention to underexplored diseases. 

## 2. Sources of ROS 

ROS are either produced through cellular processes or environmental factors. Of note, oxidative stress has been described as a secondary effect within the pathology of several rare monogenic diseases and sometimes been called a common denominator [10,11]. For example, the Friedreich’s ataxia protein FXN was shown to play a role in mitochondria biogenesis [12] and iron–sulfur cluster synthesis [13]. Via these functions, mutation of FXN leads to oxidative stress, which contributes to the pathology of Friedreich’s ataxia [14]. Such effects in which oxidative stress appears as a molecular symptom of the original genetic disruption have been extensively discussed elsewhere (e.g., [10,15,16,17]). Herein, we focus on pathogenic genetic variants that affect the main genes directly responsible for internal ROS production and hence have the immediate ability to cause oxidative imbalance in either direction and impair physiological function. 

### 2.1. The Mitochondrial Respiratory Chain

The mitochondrial respiratory chain, or electron transport chain (ETC), is one of the biggest endogenous sources of ROS in eukaryotic cells which use aerobic metabolism (Figure 1) [18,19]. In humans, the enzyme complexes I (NADH:ubiquinone oxidoreductase), II (succinate:ubiquinone oxidoreductase), and III (ubiquinol:cytochrome c oxidoreductase) are somewhat “leaky”, leading to direct one-electron transfer to molecular oxygen and the perpetual production of the superoxide anion radical (O_2·−_) as side reactions [19,20].

Complex I is encoded by more than 40 genes [21], complex II by four genes [22] and complex III by 11 genes [23]. However, the specific sites and levels of ROS production within the different protein complexes depend on the respiration substrate, whether the cell experiences norm- or hypoxic conditions, and on the inhibitor used in experimental setups [24]. Furthermore, respiratory complexes can assemble into supercomplexes, which decreases ROS production [25]. Astrocytes contain high amounts of free complex I and thus potentially higher levels of ROS compared to neurons, which display complex I and III assembly [26]. Thus, the diversity of conditions as well as of cell type specificities make it often hard to pin down the specific set of genes of respiratory complex subunits which involve pathomechanisms related to ROS production in humans. This is also exemplified by the fact that some pathogenic mutations within subunits do not lead to ROS elevation, as shown for H_2_O_2_ in a mouse model for Leigh syndrome [27]. By contrast, in other experiments, mutations of genes involved in complex I, II, or III formation were indeed shown to create greater “leakiness”, and cause elevated ROS levels and oxidative stress [21,28,29,30]. 

Since the ETC generates a strong proton gradient used for oxidative phosphorylation, transport processes, and heat production, it can be difficult to disentangle whether the pathophysiological mechanism triggered by mutation of ETC genes is based on compromised primary functions, or elevated ROS levels. However, at least for complex I deficiency, elevated ROS production and its consequent oxidative damage were shown to induce apoptotic molecular pathways leading to neuron degeneration and hence neurological symptoms, the main features of the disorder [28].

Deficiencies in respiratory complexes have similar but wide-ranging symptoms from neonatal death, lactic acidosis, myopathy, hepatopathy, encephalopathy, Leigh syndrome [31], Leber hereditary optic neuropathy (LHON) [32], to adult-onset neurological symptoms such as some forms of Parkinson disease [33,34]. Isolated complex I deficiency is the most prevalent genetic disorder of oxidative phosphorylation [35]. 

As outlined above, it is challenging to disentangle the underlying factors, but the numerous potential ROS production sites and levels might be part of the explanation of the broad range of symptoms and disorder severities. 

### 2.2. Heme in Red Blood Cells

Another ROS formation hotspot, despite being devoid of mitochondria, is the cytosol of red blood cells (RBCs). RBCs transport oxygen from the lung to peripheral tissues and superoxide can be formed when O_2_ interacts with the iron (Fe^2+^) of the heme group of hemoglobin (Hb) [36]. Spontaneous autoxidation results in O^2·−^ and methemoglobin (Hb-Fe^3+^, MetHb, Hb M) (Figure 1) [35,37], and ca. 1% of all Hb is present as Hb M in healthy individuals [38].

This normally occurring rate of superoxide production can be exceeded by pathogenic genetic variants that cause conformational changes to the globin chain proteins that contain heme as a prosthetic group. This phenomenon is well described for, e.g., autosomal recessive sickle cell disease (SCD) (Table 1). SCD is the most common severe hemoglobinopathy worldwide and is caused by a missense pathogenic variant in the globin beta-chain (*HBB*), resulting in Hb S (NM_000518.5(*HBB*):c.20A>T (p.Glu7Val); E6V) [39], which is unstable, prone to polymerization and autoxidation [40]. Besides other ROS sources like iron release, this autoxidation is the primary source for oxidative stress in sickle cells, and it leads to loss of membrane structure and function and consequent multisystem disease [41,42,43,44]. When iron is released, it impacts the oxidative balance through transfer of single electrons via the Fenton reaction (Fe^2+^ + H_2_O_2_ → Fe^3+^ + HO**^·^** + OH^−^) and the Haber–Weiss reaction (O_2_**^·^**^−^ + H_2_O_2_ → O_2_ + HO**^·^** + OH^−^ catalyzed by iron) (Figure 1) [45].

Also, other *HBB* variants have been described to cause methemoglobinemia. For example, the so-called Hb M-Hyde Park (Hb M-Akita) (NM_000518.4(*HBB*):c.277C>T (p.His93Tyr)) variant leads to conformational changes at the heme binding site and hence higher rates of methemoglobin formation. Inheritance is autosomal dominant [38,46]. 

### 2.3. NADPH Oxidases and Myeloperoxidase

Besides these cellular processes in which ROS are formed passively as byproducts, the cell can actively produce high levels of ROS through dedicated enzymes. NADPH oxidases (NOX) use reduced nicotinamide adenine dinucleotide phosphate (NADPH) to generate superoxide (Figure 1) [47]. NOX play an important role in the innate host defense. For the so-called “respiratory burst”, superoxide is released into the extracellular space or phagosomes to fight off pathogenic bacteria or fungi [48]. Furthermore, NOX enzymes were found to produce intracellular ROS at lower levels, which is believed to serve signaling functions [49] and to control the cellular redox balance by oxidizing NADPH and through ROS formation [50]. The importance of the ability to generate high ROS levels is exemplified by the detrimental consequences of NOX disruption.

Several monogenic disorders are related to NOX subunits. The genes that encode the NOX2 complex in phagocytes are related to chronic granulomatous disease [51,52]: *CYBB* pathogenic variants cause X-linked recessive chronic granulomatous disease (CGDX, [53]) and immunodeficiency 34 [54]), *CYBA* pathogenic variants cause autosomal recessive (AR) CGD4 [55], *NCF1* pathogenic variants cause AR CGD1 [56], *NCF2* pathogenic variants cause AR CGD2 [57], and *NCF4* pathogenic variants cause AR CGD3 [58] (Table 1). These rare primary immunodeficiencies increase susceptibility to life-threatening bacterial and fungal infections and lead to development of granulomas [59]. 

Pathogenic variants in other NADPH oxidases were shown to increase susceptibility to inflammatory bowel disease (*NOX1* and *DUOX2*, [60]), or cause congenital hypothyroidism (*DUOX2*, Thyroid dyshormonogenesis 6, AR, [61]; *DUOXA2*, Thyroid dyshormonogenesis 5, AR, [62], Table 1). The latter one is the result of disrupted H_2_O_2_ production through mutated DUOX2, which would be required for organification of iodide for thyroid hormone synthesis catalyzed by thyroid peroxidase [52,63].

Another enzyme of the innate immune response is the heme-containing enzyme myeloperoxidase (MPO, Table 1, Figure 1). It is highly abundant in azurophilic granules of neutrophils [64] and is a crucial component in neutrophil extracellular nets [65]; it is also found—to a lesser extent—in monocytes [66]. To kill pathogens, MPO creates strongly reactive species, such as hypochlorous acid (HOCl^.^) from hydrogen peroxide (H_2_O_2_) and chloride (Cl^−^) [67,68]. Autosomal recessive MPO deficiency is the most common phagocyte defect but is asymptomatic in the majority of patients, suggesting compensatory mechanisms [69]. However, higher susceptibility to fungal infections, especially *Candida albicans*, has been described [70]. This vulnerability might become apparent only in combination with comorbidities like diabetes, which itself increases the risk of infections [68]. Similarly, a recent study described a case of partial DiGeorge syndrome together with MPO deficiency. DiGeorge syndrome is characterized by immunodeficiency among other symptoms. However, the patient had more frequent and severe infections than expected for partial DiGeorge alone, which the authors explained by co-occurrence of MPO deficiency [71]. On the other hand, a protective mechanism against cardiovascular disease by absence of the potentially oxidative stress causing MPO enzyme has also been discussed [69]. 

### 2.4. Other ROS Sources 

In addition to the above-described reactions and enzymes, there are additional sources that can produce significant amounts of reactive species under certain physiological circumstances. 

For example, xanthine oxidoreductase (XDH/XOD), encoded by the *XDH* gene (Table 1, Figure 1), exists in two interconvertible isoforms. Both forms utilize hypoxanthine or xanthine to produce uric acid. However, the predominant form has xanthine dehydrogenase (XDH) activity, and uses NAD^+^ as cofactor to produce NADH, whereas the xanthine oxidase (XOD) form uses oxygen to produce the superoxide anion and H_2_O_2_ [72]. XDH can be converted to XOD by irreversible limited proteolysis or reversibly by thiol oxidation (reviewed in [73]), e.g., in hypoxic/ischemic tissue [74]. Homozygous or compound heterozygous pathogenic *XDH* variants cause type I xanthinuria (XAN1) [75]. The symptoms, low serum and urine uric acid and xanthinuria leading to urolithiasis [76], result from XDH’s primary function. Mutations that lead to elevated ROS production are not described for this enzyme to the best of our knowledge. 

Furthermore, nitric oxide synthases (NOS1-3) can become sources of O_2_**^·^**^−^ (Figure 1). Normally, they use L-arginine to produce nitric oxide (·NO), which belongs to the group of reactive nitrogen species (RNS) and serves as an important signaling molecule, especially for the vascular tone [77]. However, persistent oxidative stress, and thus reduced levels of the cofactor (6R)-5,6,7,8-tetrahydro-L-biopterin (BH_4_), lead to uncoupling of endothelial NOS (eNOS, NOS3) so that the enzyme produces O_2_**^·^**^−^ instead of ·NO [78]. Furthermore, O_2_**^·^**^−^ is produced by NOS2 in arginine-depleted macrophages [79]. Although some *NOS3* variants have been described to increase the risk for certain conditions like pregnancy- induced hypertension [80] or ischemic stroke [81], no clear monogenic disorder has been described for *NOS1*, *NOS2*, or *NOS3*.

At lower amounts, ROS can also be byproducts of other enzymes like cytochrome P450 or cyclooxygenase (reviewed in [82]). 

Environmental factors that create elevated ROS levels are, e.g., UV radiation, ionizing radiation, smoking and air pollution, chemicals such as drugs, and certain types of food like the fava bean [2,83]. Often, genetic defects in the anti-oxidative machinery remain unnoticed until such environmental stresses hit and overwhelm the residual anti-oxidative capacity of the cell, e.g., in hemolytic anemia due to glucose 6-phosphate dehydrogenase (G6PDH) deficiency (see below).

## 3. The Anti-Oxidative Defense System

Protective mechanisms have evolved that keep the redox balance in check or clear ROS-induced damage. There are several ways to categorize the different parts of the overall cellular anti-oxidative defense system. Ighodaro and Akinloye [84] propose to categorize the individual parts of the anti-oxidative machinery into first, second, third, and fourth line defense. 

First line antioxidants prevent or reduce ROS formation. The most important agents in this category are superoxide dismutase (SOD), catalase (CAT), glutathione peroxidase (GPX), and peroxiredoxins (PRDX). SOD dismutates the superoxide anion O^2·−^ to H_2_O_2_, and CAT detoxifies H_2_O_2_ to water and oxygen. GPX and PRDX convert H_2_O_2_ to water, are oxidized themselves, and need to be reduced again to stay functional (Figure 2, [3]). Second-line anti-oxidants act as so-called “scavengers” and prevent ROS from damaging biomolecules or forming further ROS. These are either small, non-enzyme molecules that are produced by human cells, like glutathione (GSH), uric acid, and ubiquinol, or proteins with thiol groups like thioredoxins (TXN) or peroxiredoxins [85,86]. Also, nitric oxide (·NO) falls into this category as it scavenges O_2_**^·^**^−^ to form peroxynitrite (ONOO^−^), which itself is highly reactive at physiological pH and needs to be further converted, e.g., by GSH [87]. Scavengers can also be small molecules taken up by nutrition, like ascorbic acid (vitamin C) and alpha-tocopherol (vitamin E). Third-line anti-oxidants repair already damaged structures, and fourth-line anti-oxidants build up prevention and adaptation mechanisms, e.g., upregulate expression of relevant genes [84] or, even faster, metabolic switches like the inactivation of glyceraldehyde 3-phosphate dehydrogenase (GAPDH), which leads to metabolic rerouting into the pentose phosphate pathway and increased NADPH production [88,89]. 

First- and second-line defense can be summarized as primary anti-oxidative defense, which serves for immediate protection [90]. Within the primary antioxidative defense, the thiols GSH and TXN play crucial roles. The tripeptide GSH scavenges ROS or, like the polypeptide TXN, serves as an electron donor for peroxidases. The peroxidases either detoxify H_2_O_2_ (GPX) or reduce oxidized cysteine residues and cleave ROS-caused disulfide bonds in proteins. In order to maintain a constant pool of reduced GSH and TXN, respective reductases recycle GSH and TXN under consumption of NADPH (Figure 2) [91,92]. 

### 3.1. Enzymes of the ROS Scavenger System

#### 3.1.1. Superoxide Dismutases

Three genes in humans (*SOD1*, *SOD2*, *SOD3*) encode for superoxide dismutases, of which only *SOD1* is described as disease associated. Amyotrophic lateral sclerosis (ALS) is a heterogenous, late-onset neurodegenerative disorder for which up to 95% of cases are sporadic. However, up to 10% of cases seem to have familial predisposition, and *SOD1* is a top hit among 40 associated genes. It is not unequivocally defined whether *SOD1* variants are causative for ALS, co-causative or modifying [93], but hypothetical pathomechanisms explain the disease with mostly heterozygous gain-of-function variants that lead to the accumulation of aggregated SOD1 protein [94,95]. Elevated oxidative stress was also described in ALS, but it is not clear whether both mechanisms are linked [96].

Interestingly, more recently, homozygous *SOD1* variants were detected in children with progressive severe developmental delay, axial hypotonia, and spastic tetraplegia in at least three families. Biochemical testing confirmed total absence of enzyme activity in the patients and 50% of enzyme activity in the parents who were heterozygous for the respective familial variant [97,98,99]. 

#### 3.1.2. Catalase

Once SOD has converted O^2^^·−^ into H_2_O_2_, the latter needs to be further detoxified. This task is efficiently accomplished by catalase that is encoded by the *CAT* gene, and which can convert millions of hydrogen peroxide molecules to water and oxygen per second [100]. Homozygous pathogenic *CAT* variants are associated with the rare condition acatalasemia, also known as acatalasia, which leads to total or near total loss of CAT activity in erythrocytes [101]. Probands with heterozygous variants are called hypocatalasemic and have blood catalase activity of about 50%. The disorder was coincidentally discovered by a Japanese doctor named Takahara in 1946, who performed oral surgery on a girl and used H_2_O_2_ to disinfect the wound. He noticed the absence of oxygen bubbles that would normally be formed due to CAT activity. Additionally, the blood turned black [102]. A dark chocolate brown color of arterial blood refers to methemoglobinemia [103]—the aforementioned phenomenon when some of the ferrous heme iron [Fe^2+^] is converted to ferric [Fe^3+^] iron in hemoglobin and hence oxygen transport is impaired. Takahara also described gangrenes, oral ulceration, and gingival necrosis in CAT-deficient patients. In the 1940s and 50s, oral hygiene was still poor in Japan and the symptoms were attributed to hydrogen peroxide produced by phagocytic cells and bacterial infections. When oral hygiene improved, these symptoms were seen less often. But besides these, the disorder used to be considered asymptomatic [104]. However, animal models displayed a lower hydrogen peroxide removal rate and elevated oxidative stress. In fact, CAT-deficient patients develop diabetes more often and earlier, and it is known that insulin-producing pancreatic beta cells are especially vulnerable to oxidative damage. Indeed, acatalasemia and hypocatalasemic patients are prone to age- and oxidative stress-related diseases such as type II diabetes, tumors, Parkinson’s disease, vitiligo, schizophrenia, or loss of soft tissues (reviewed in [105,106]). 

#### 3.1.3. Glutathione Peroxidases

Glutathione peroxidases (GSX) are a family of oxidoreductases that reduce H_2_O_2_, organic hydroperoxides and lipid peroxides by using GSH as a reductant. In humans, eight GPX enzymes are known to be encoded by the *GPX1*, *GPX2*, *GPX3*, *GPX4*, *GPX5*, *GPX6*, *GPX7,* and *GPX8* genes [107]. Of these, *GPX1* and *GPX4* were mentioned in association with rare monogenic disorders. 

Biallelic pathogenic variants of *GPX1* have been described as associated with hemolytic anemia [108]. Hemolysis due to lipid peroxidation is a well-described consequence of oxidative stress [109]. However, a clear genotype–phenotype relationship could not be unequivocally established to date. 

Biallelic pathogenic *GPX4* variants cause the rare and fatal disorder Sedaghatian type of spondylometaphyseal dysplasia (SSMD, [110]), with only 24 cases described to date. SSMD is a severe metaphyseal chondrodysplasia with, e.g., delayed epiphyseal ossification, mild limb shortening, platyspondyly, pulmonary hemorrhage, and brain anomalies resulting in death due to respiratory failure or cardiac abnormalities in the perinatal period [111]. Disruption of GPX4 causes buildup of lipid peroxidation and ferroptosis, which is an iron-dependent, non-apoptotic form of cell death [112,113]. 

#### 3.1.4. Peroxiredoxins

Peroxiredoxins are ubiquitous cysteine-containing proteins that are highly abundant, especially in RBCs, and play crucial roles in regulating intracellular peroxide levels [114]. In humans, there are six genes: *PRDX1*, *PRDX2*, *PRDX3*, *PRDX4*, *PRDX5,* and *PRDX6*. Of these, *PRDX1*, *PRDX2* and *PRDX4*-*6* have not been described as related to a monogenic disorder. 

To the best of our knowledge, thus far, only *PRDX3*, which is localized to mitochondria [115], is associated with monogenic disorders. In only eight cases, and rather recently, homozygous pathogenic *PRDX3* variants have been described to be causative for spinocerebellar ataxia-32 (SCAR32) [116,117,118]. Patients presented with gait ataxia, cerebellar ataxia, upper limb ataxia, dysarthria, dysphagia, oculomotor signs, sometimes hyper- or hypokinetic movement abnormalities, and cerebellar atrophy in one patient. The disease mechanism was shown to be oxidative stress and mitochondrial damage due to absent or nearly absent expression of PRDX3 protein [116,118]. 

Heterozygous pathogenic *PRDX3* variants have been described in segregation studies as related to punctiform and polychromatic pre-Descemet corneal dystrophy (PPPCD). The phenotype, which was not reported in patients with homozygous pathogenic *PRDX3* variants, is characterized by hyperreflective deposits in the posterior stroma of the cornea of the eye. Affected individuals are asymptomatic with no visual impairment [119,120,121].

*PRDX1* together with the *MMACHC* gene have been described as associated with the digenic disorder autosomal recessive combined methylmalonic aciduria and homocystinuria type cblC. *PRDX1* participates in the disease pathology by affecting *MMACHC* expression [122]. 

### 3.2. Electron Donors

#### 3.2.1. Glutathione 

GSH is a small tripeptide composed of cysteine, glutamate, and glycine. Due to the cysteine’s thiol group, two GSH molecules can form a disulfide bond and thus serve as electron donors. GSH synthesis is catalyzed by the enzymes glutamate–cysteine ligase (*GCLC*, catalytic subunit, γ-glutamylcysteine synthetase) and glutathione synthetase (*GSS*), which are part of the γ-glutamyl cycle [123]. Pathogenic variants in both genes are described to be related to ultra-rare metabolic disorders. 

Glutamate–cysteine ligase deficiency is an autosomal recessive disorder that is characterized by hemolytic anemia and variable other features like myopathy or late-onset spinocerebellar degeneration [124]. Only very few patients have been described to date: five until 1996 and none more recently to our knowledge. The GSH content was very low in the RBCs of all five patients (between 2 and 13.1% of normal) [125]. 

For glutathione synthetase deficiency, the OMIM database distinguishes between two forms: isolated hemolytic anemia [126] and a severe, multi-tissue form characterized by hemolytic anemia, central nervous system symptoms (e.g., mental retardation, ataxia), metabolic acidosis and severe urinary excretion of 5-oxoproline [127]. In the latter case, different cell types have severely reduced levels of enzyme activity and GSH content, whereas nucleated cells are able to maintain substantial levels of GSH in patients with isolated hemolytic anemia [128]. It was proposed that accumulation of the GSS substrate γ-glutamylcysteine substitutes GSH at least partially as an antioxidant [129]. 

No null alleles have been described for both disorders; and complete absence of GSH due to *Gclc* or *Gss* knockout is lethal in mice [130,131,132]. 

#### 3.2.2. Thioredoxins

Thioredoxins are small proteins that carry neighboring thiol groups in their active site that can form a disulfide bond and thereby serve as electron donors. Two thioredoxin genes have been described in humans: *TXN* (cytosolic) and *TXN2* (mitochondrial). *Txn* as well as *Txn2* knockout is embryonically lethal in mice [133,134]. 

While no genetic disorder has been described for *TXN,* the OMIM database mentions a phenotype named “Combined oxidative phosphorylation deficiency 29” (Table 1) for *TXN2*. So far, only one patient with absent TXN2 protein in fibroblasts, disrupted ROS homeostasis, and impaired mitochondrial ATP generation due to a homozygous *TXN2* loss-of-function variant was identified by whole exome sequencing [135]. These cellular phenotypes could be rescued by lentiviral expression of the wild-type TXN2 protein, and supplementation of antioxidants had beneficial effects on the patient’s condition in short-term follow-up. The 16-year-old patient’s main symptoms were microcephaly, optic atrophy, abnormal muscle tone, global developmental delay, seizures, cerebellar atrophy, spasticity, and increased serum lactate at the time [135]. Nevertheless, the case remains interesting as no other pathogenic or likely pathogenic variant has been described for *TXN2*, and loss of TXN has been described as lethal in mice, which points to compensatory mechanisms in the patient described by Holzerova et al. [135].

#### 3.2.3. Reductases for Recycling of Electron Donors

Under constant ROS production, the pools of anti-oxidant electron donors would quickly deplete. Hence, reductases recycle the reduced forms of glutathione and thioredoxin, utilizing NADPH. 

##### Glutathione Reductase, GSR

Glutathione reductase is encoded by only one gene in humans (*GSR*)*,* or mice (*Gsr*). Knockout of *Gsr* in mice results in a healthy phenotype [136,137]. However, homozygous pathogenic *GSR* variants in humans result in hemolytic anemia and cataract or hyperbilirubinemia and jaundice (Table 1) [138]. GSR activity was very low or absent in tested cells in patients from two families [139,140]. Normal glucose 6-phosphate dehydrogenase (G6PDH) activity, whose disruption could result in similar symptoms, was shown in the infant with jaundice described by Kamerbeek et al. [140]. However, in the few studies available, no other possible genetic causes for the described symptoms were genetically excluded, e.g., by whole exome analysis. That humans can be rather robust toward reduced GSR activity is also exemplified by the high frequency of moderate GSR deficiency due to malnutrition, specifically due to the lack of riboflavin, which is a precursor of the GSR cofactor flavin adenine dinucleotide (FAD) in some populations [141,142]. 

##### Thioredoxin Reductase, TXNR

Humans express three thioredoxin reductase isoforms: TXNRD1, TXNRD2, TXNRD3. *Txnrd1* and *Txnrd2* knockout were shown to be embryonically lethal in mice [143]. *Txnrd3* knockout mice are viable and without discernible phenotype. However, male mice have impaired fertility, which points to a rather specific function of Txnrd3 in spermatogenesis [144]. In humans, to our knowledge, no disease phenotype has been described for all three genes.

#### 3.2.4. NADPH—The Key Coenzyme of the Anti-Oxidant Response

Regeneration of reduced thiols from disulfides by reductases requires a constant supply of the universal redox coenzyme NADPH. Several cytosolic and mitochondrial enzymes convert NADP^+^ into NADPH. 

##### The Oxidative Pentose Phosphate Pathway, oxPPP

The most prominent enzyme in the category of NADPH production is the pentose phosphate pathway enzyme glucose 6-phosphate dehydrogenase (G6PD). G6PD is generally considered to be the main source of cytoplasmic NADPH [145]. Complete G6PD knockout was shown to be embryonically lethal in mice [146]. In humans, more than 200 pathogenic *G6PD* variants have been described. The most severe manifestations have residual enzyme activity of around 5% [147]. With over 400 million cases worldwide, X-linked recessive G6PD deficiency (Table 1) is the most common enzymopathy in humans, and the most common symptoms are neonatal jaundice and acute hemolytic anemia [148]. Hemolysis is mainly triggered by external factors such as drugs, certain foods (e.g., fava beans), or infection [149]. Otherwise, individuals with G6PD deficiency are mostly symptom-free [148]. The high frequencies of G6PD alleles in populations that geographically overlap with the distribution of malaria-causing *Plasmodium falciparum* indicate at least partial protective effects of these alleles against the infectious disease [147]. G6PD deficiency is an example of a redox disorder that only displays symptoms when the anti-oxidative capacity of the cell is overwhelmed due to acute oxidative stress and insufficient NADPH supplies. Otherwise, the residual activity of G6PD is sufficient to maintain health. Additionally, it was shown that, although G6PD is a major source of NADPH, it is dispensable for pentose production [150]. 

The second NADPH-producing enzyme of the oxPPP is 6-phosphogluconate dehydrogenase (PGD). PGD deficiency (Table 1) was found to cause transient hemolysis and well-compensated chronic nonspherocytic hemolytic anemia. Inheritance was described as autosomal dominant [151]. However, the genotype-phenotype relationship is not unequivocally established and individuals who were found in population surveys who had PGD activity of less than 5% in RBC were completely healthy [152].

##### Malic Enzyme and Glutamate Dehydrogenase

There are two NADPH-producing isoforms of malic enzyme (ME) in humans: cytosolic ME1 and mitochondrial ME3. ME1 and 3 catalyze the reversible decarboxylation of malate to pyruvate. In mice, knockout of *Me1* causes decreased body weight [153,154], and no specific phenotype was published for knockout of *Me3* to our knowledge. Neither of the two genes has been described as associated with monogenic disorders in humans. 

Glutamate dehydrogenase (GLUD) catalyzes the conversion of glutamate to α-ketoglutarate and ammonia while producing NADPH [155]. There are two mitochondrial isoforms, GLUD1 and GLUD2. Only *GLUD1* has been described in association with a genetic phenotype—autosomal dominant hyperinsulinism-hyperammonemia syndrome (Table 1) [156]. 

##### Isocitrate Dehydrogenase

Humans possess two IDH isoforms that produce NADPH by converting isocitrate to α-ketoglutarate (α-KG): cytosolic IDH1 and mitochondrial IDH2. *Idh1* knockout is viable in mice, but the animals display greater susceptibility to oxidative stress [157]. No monogenic disorder has been described for *IDH1* in humans, although pathogenic IDH1 variants are common in human neoplastic disorders [158]. *Idh2* knockout mice show susceptibility to mitochondrial dysfunction, cardiac hypertrophy, and accelerated age-related hearing loss [159,160]. However, in humans, *IDH2* has been described as related to D-2-hydroxyglutaric aciduria 2 (D2HGA2, Table 1). Kranendijk et al. identified 15 unrelated patients with specific de novo heterozygous variants affecting codon R140 (NM_002168.2). D2HGA2 has a wide phenotypic spectrum from asymptomatic to severe neurological disorder with developmental delay, epilepsy, dysmorphic features, and cardiomyopathy. The phenotype was not described to be caused by elevated oxidative stress but by an increase in a promiscuous newly introduced enzymatic function that is of low activity in the wild type enzyme: the described variants reduce the enzyme’s ability to convert isocitrate to α-KG but increase the conversion of α-KG into D-2-hydroxyglutarate [158]. 

##### Folate Metabolism

The importance and amount of NADPH derived from folate metabolism have been described in several publications in 2014 [145,161,162]. In proliferating cells—hence not in RBCs—the NADPH production from folate metabolism almost equals that from the oxidative pentose phosphate pathway [161]. 

The folate cycle gene methylenetetrahydrofolate dehydrogenase (MTHFD) converts 5,10-methylenetetrahydrofolate and NADP^+^ into 5,10-methenyltetrahydrofolate and NADPH [163]. Knockout of the genes *Mthfd1* or *Mthfd2* is embryonically lethal in mice [164,165]. No genetic disease has been described for the human mitochondrial isoform MTHFD2. However, autosomal recessive pathogenic *MTHFD1* variants cause the ultra-rare disorder “combined immunodeficiency and megaloblastic anemia with or without hyperhomocysteinemia” (Table 1) [166,167]. Functional studies on patient fibroblasts revealed severely impaired methionine production but did not analyze for potential oxidative stress in these cells [168]. 

The folate cycle enzyme formyltetrahydrofolate dehydrogenase, alternatively named aldehyde dehydrogenase 1 family member L (ALDH1L), produces tetrahydrofolate and CO_2_ from 10-formyltetrahydrofolate under simultaneous NADPH production [169]. There are two isoforms in humans: cytosolic ALDH1L1 and mitochondrial ALDH1L2 [170]. Knockout of *Aldh1l1* in mice resulted in no obvious phenotype [171], and no inherited monogenic disorder has been described in humans. Interestingly, for *ALDH1L2*, one patient was described with a neuro-ichthyotic syndrome characterized by ichthyosis, thick fingers and toes, dysmorphic features, and developmental delay. Two *ALDH1L2* variants were detected in the index patient in compound heterozygosity, which indicates autosomal recessive inheritance. Functional analyses in the patient’s fibroblasts confirmed impaired mitochondrial metabolism with impact on β-oxidation of fatty acids [172]. Knockout of *Aldh1l2* resulted in normally appearing mice. However, in-depth physiological analysis confirmed, e.g., reduced NADPH, abnormal lipid metabolism, showed drastically reduced GSH levels, and increased oxidative stress which impacted β-oxidation of fatty acids [173]. Nevertheless, further patients from different families would support an unequivocal genotype-phenotype association. 

**Table 1 biomolecules-14-00206-t001:** Summary of discussed genes and phenotypes. AR, autosomal recessive, AD, autosomal dominant, XLR, X-linked recessive, OMIM**^®^**, Online Mendelian Inheritance in Man database.

Protein	Gene (OMIM^®^ no.)	Monogenic Disease (OMIM^®^no^.^)	Inheritance	Described by
ROS production				
Hemoglobin beta-locus	*HBB* (141900)	Sickle cell disease (603903)	AR	[39]
		Methemoglobinemia, beta type (617971)	AD	[38]
Cytochrome b(-245), beta subunit, p91-phox	*CYBB* (300481)	Chronic granulomatous disease (306400)	XLR	[53]
		Immunodeficiency 34, mycobacteriosis (300645)	XLR	[54]
Cytochrome b(-245), alpha subunit, p22-phox	*CYBA* (608508)	Chronic granulomatous disease 4 (233690)	AR	[55]
Neutrophil cytosolic factor-1, p47-phox	*NCF1* (608512)	Chronic granulomatous disease 1 (233700)	AR	[56]
Neutrophil cytosolic factor-2, p67-phox	*NCF2* (608515)	Chronic granulomatous disease 2 (233710)	AR	[57]
Neutrophil cytosolic factor-4, p40-phox	*NCF4* (601488)	Chronic granulomatous disease 3 (613960)	AR	[58]
Myeloperoxidase	*MPO* (606989)	Myeloperoxidase deficiency (254600)	AR	[70]
Xanthine dehydrogenase	*XDH* (607633)	Xanthinuria, type I (278300)	AR	[75]
Nitric oxide synthase 1	*NOS1* (163731)	-		
Nitric oxide synthase 2	*NOS2* (163730)	-		
Nitric oxide synthase 3	*NOS3* (163729)	-		
enzymatic ROS clearance				
superoxide dismutase 1	*SOD1* (147450)	Amyotrophic lateral sclerosis 1 (105400)	AD, AR	[93]
		Spastic tetraplegia and axial hypotonia, progressive (618598)	AR	[99]
superoxide dismutase 2	*SOD2* (147460)	-		
superoxide dismutase 3	*SOD3* (185490)	-		
catalase	*CAT* (115500)	Acatalasemia (614097)	AR	[102]
glutathione peroxidase 1	*GPX1* (138320)	Hemolytic anemia due to glutathione peroxidase deficiency (614164)	AR	[108]
glutathione peroxidase 4	*GPX4* (138322)	Spondylometaphyseal dysplasia, Sedaghatian type (250220)	AR	[110]
glutathione peroxidase 2, 3, 5-8	*GPX2* (138319) *GPX3* (138321) *GPX5* (603435) *GPX6* (607913) *GPX7* (615784) *GPX8* (617172)	-		
peroxiredoxin 3	*PRDX3* (604769)	Corneal dystrophy, punctiform, and polychromatic pre-Descemet (619871)	AD	[120]
		Spinocerebellar ataxia, autosomal recessive 32 (619862)	AR	[116]
peroxiredoxin 1, 2, 4, 5, 6	*PRDX1* (176763) *PRDX2* (600538) *PRDX4* (300927) *PRDX5* (606583) *PRDX6* (602316)	-		
thiols and thiol production				
gamma-glutamylcysteine synthetase, catalytic subunit	*GCLC* (606857)	Hemolytic anemia due to gamma-glutamylcysteine synthetase deficiency (230450)	AR	[124]
glutathione synthetase	*GSS* (601002)	Glutathione synthetase deficiency (266130)	AR	[127]
		Hemolytic anemia due to glutathione synthetase deficiency (231900)	AR	[126]
thioredoxin 2	*TXN2* (609063)	Combined oxidative phosphorylation deficiency 29 (616811)	AR	[135]
thioredoxin	*TXN* (187700)	-		
thiol recycling				
glutathione reductase	*GSR* (138300)	Hemolytic anemia due to glutathione reductase deficiency (618660)	AR	[138]
thioredoxin reductase 1, 2, 3	*TXNRD1* (601112) *TXNRD2* (606448) *TXNRD3* (606235)	-		
NADPH production				
glucose 6P-dehydrogenase	*G6PD* (305900)	Hemolytic anemia, G6PD deficient (favism) (611162)	XL	[147]
6-phosphogluconate dehydrogenase	*PGD* (172200)	Phosphogluconate dehydrogenase deficiency (619199)	AD	[151]
malic enzyme 1	*ME1* (154250)	-		
malic enzyme 3	*ME3* (604626)	-		
glutamate dehydrogenase 1	*GLUD1* (138130)	Hyperinsulinism-hyperammonemia syndrome (606762)	AD	[156]
glutamate dehydrogenase 2	*GLUD2* (300144)	-		
isocitrate dehydrogenase 2	*IDH2* (147650)	D-2-hydroxyglutaric aciduria 2 (613657)	AD	[158]
isocitrate dehydrogenase 1	*IDH1* (147700)	-		
methylenetetrahydrofolate dehydrogenase 1	*MTHFD1* (172460)	Combined immunodeficiency and megaloblastic anemia with or without hyperhomocysteinemia (617780)	AR	[166]
methylenetetrahydrofolate dehydrogenase 2	*MTHFD2* (604887)	-		
10-formyltetrahydrofolate dehydrogenase (mitochondrial)	*ALDH1L2* (613584)	neuro-ichthyotic syndrome?	AR	[172]
10-formyltetrahydrofolate dehydrogenase (cytosolic)	*ALDH1L1* (600249)	-		

## 4. Conclusions

In this review, we explored monogenic diseases related to genes associated with endogenous ROS production and anti-oxidative defense systems, and we described the most common as well as some of the rarest genetic disorders. Of note, this article aimed at providing an overview of the most significant genes involved in redox metabolism and its regulation—while there are numerous further metabolic pathway enzymes located upstream of the described genes that could also have a potential impact on the discussed functions. Herein, attention was drawn to extremely rare genetic conditions. For some disorders, only few patients have been described despite the recent surge in sample throughput by more and more affordable sequencing technologies, as well as fast progress in other analytical technologies, including metabolomics and proteomics. Since a vast number of samples are processed by diagnostic companies and not research active clinics or institutes, comprehensive re-analysis of proprietary databases might identify further patients that would back up weak genotype-phenotype links. This would not only improve diagnostics of rare diseases but also provide valuable information for basic biochemical research. 

## Figures and Tables

**Figure 1 biomolecules-14-00206-f001:**
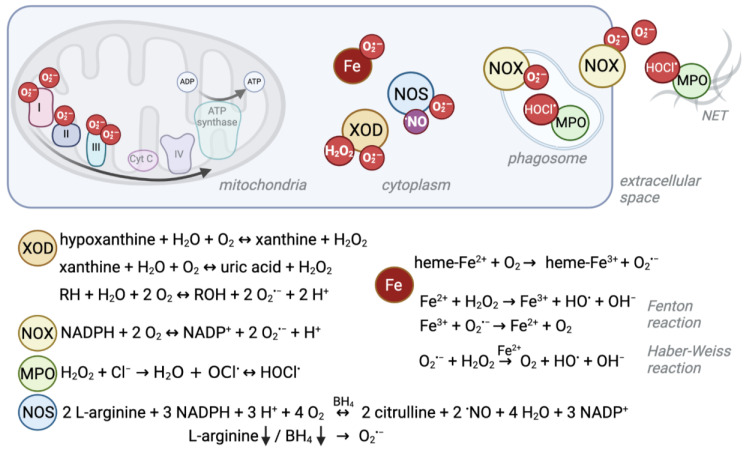
Schematic overview of ROS production sites and reactions. Several enzymatic reactions as well as interactions of iron with oxygen give rise to reactive oxygen species. Their localization and production level can differ between cell compartments and cell types. For example, red blood cells experience high levels of superoxide due to their heme-bound iron (Fe^2^+) and oxygen interaction, whereas the mitochondrial electron transport chain is a major source of ROS in other cells. Under ischemic conditions, xanthine oxidase (XOD) produces superoxide anions (O_2_**^·^**^−^) and hydrogen peroxide (H_2_O_2_) in the cytosol. NADPH oxidases (NOX) mainly produce O_2_**^·^**^−^ to kill pathogens in the phagosome or extracellular space. Similarly, myeloperoxidase (MPO) produces hypochlorous acid (HOCl^.^) from H_2_O_2_ which derives from NOX-produced O_2_^·−^ in phagosomes and neutrophil extracellular traps (NETs). Instead, nitric oxide synthases (NOS) normally produce nitric oxide (^.^NO) as a signaling molecule. Depletion of its substrate arginine or cofactor (6R)-5,6,7,8-tetrahydro-L-biopterin (BH_4_) can cause the enzyme to uncouple and produce O_2_**^·^**^−^.

**Figure 2 biomolecules-14-00206-f002:**
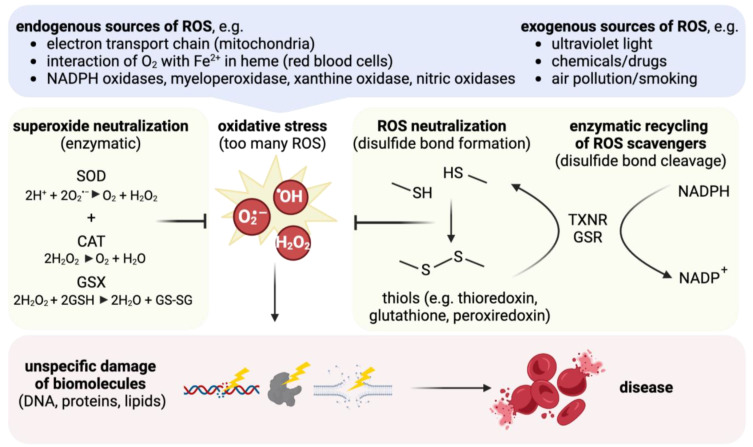
The cellular anti-oxidative machinery. Oxidative stress from exogenous or endogenous sources would damage biomolecules and cause disease if not counteracted by a set of direct enzymatic (SOD, CAT, GSX) and non-enzymatic (GSH, TXN) anti-oxidative molecular systems that are regenerated by different reductases (TXNR, GSR).

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
