# Peer review of "Monogenic Disorders of ROS Production and the Primary Anti-Oxidative Defense"

_biomolecules, 2024, doi:10.3390/biom14020206_

Round 1

Reviewer 1 Report

Comments and Suggestions for Authors

The manuscript entitled „Monogenic Disorders of ROS Production and the Primary Anti-Oxidative Defense” provides an interesting overview of the ROS generation and/or detoxification, and the genetic disorders that lead to dysregulation of these vital processes.

The article proposed by the authors is represented by a systematization of literature data on monogenic diseases related to genes associated with endogenous ROS production and anti-oxidative defense systems.

I agree with the Authors that research on specific metabolic pathways located on the genes described in the review, related to the occurrence of oxidative stress, could allow the description of genotype-phenotype correlations in the course of numerous diseases, including rare diseases.

In the attached table, the Authors present a systematic summary of the discussed genes and phenotypes, taking into account specific disease entities.

The review is focused, timely interesting and substantially detailed. I recommend publication of the work because, in my opinion, the manuscript is interesting for the Readers.

Author Response

Dear Sir or Madam,

thank you very much for your feedback for our manuscript „Monogenic Disorders of ROS Production and the Primary Anti-Oxidative Defense” and considering our manuscript interesting for potential readers.

We made some additions, especially in the "ROS production section" as requested by another reviewer. There is now also a second figure included which provides an overview of the mentioned ROS production sites and reactions.

All changes are highlighted in yellow.

Thank you once more and kind regards,

Nana-Maria Grüning

Reviewer 2 Report

Comments and Suggestions for Authors

The manuscript of Gruning et al. is aimed at providing a comprehensive review concerning the current knowledge of monogenic disorders of the ROS producing and eliminating systems.  Authors divide their manuscript into the following chapters: Introduction, ROS sources, Anti-oxidative defense systems and Conclusion. The chapters “ROS sources” and “Anti-oxidant defense systems” comprise of several sub-chapters listing ROS sources and anti-oxidant molecules/enzymes, respectively. Authors complement their manuscript with 1 Figure that illustrates the cellular anti-oxidant systems and a Table that summarizes the monogenic disorders discussed in the manuscript.

The theme is original and current, and of interest for the readers of Biomolecules. The text is well written, easy to follow with a logical thought-flow and provides a balanced review of current knowledge. Authors cite a vast list of 153 references to provide a complete overview.

This is a quality review and this reviewer has further suggestions.  

Author Response

(The authors gave the same response as above.)

Reviewer 3 Report

Comments and Suggestions for Authors

This is an overall well-written review article summarizing the monogenic metabolic diseases related to redox metabolism, ROS production and how the anti-oxidative defense system could be affected. I only suggest to include, in Table 1, a column with specific references for the reported variants associated to the corresponding disease. Also, a mechanistic figure summarizing enzymes and their networks, ROS products and subcellular localization would help the reader to understand the relationship between them and their potential role in disease.

Comments on the Quality of English Language

I will suggest a careful review of the different sections of the manuscript for typos.

Author Response

Dear Sir or Madam,

thank you very much for your feedback for our manuscript „Monogenic Disorders of ROS Production and the Primary Anti-Oxidative Defense” and considering our manuscript interesting for potential readers.

I included a column in Tbl. 1 with a reference to the respective disorder, as suggested by you. 

You also asked for another figure "Also, a mechanistic figure summarizing enzymes and their networks, ROS products and subcellular localization would help the reader to understand the relationship between them and their potential role in disease."

I totally understand the value for such a figure and made several attempts of drafting one, even for the first submission. Hence, it made me happy that you came to the same conclusion that such a figure would fit the manuscript.

However, the complexity of the topic made it impossible for me to come up with something clear and correct, especially in the given timeframe. I would love a figure that displays the different ROS production sites together with the enzymes and factors of the anti-oxidative machinery distributed onto their respective compartments. However, for many enzymes there are not even unambiguous data available in which cell types or compartments they are expressed, or there is data from mouse/rat which I couldn't use in an article about human diseases. For example, glutathione peroxidase has 8 isoforms. GPX4 is associated with membranes, but for some other isoforms, I wouldn't be sure where to put them. At some point I felt the risk of introducing false or yet unestablished claims would be too high or the figure would be too schematic and then not that useful anymore.

As a compromise - and I think also to the benefit of the manuscript - I included an overview figure of the ROS production sites, which is a subset of the information of an overall overview figure, together with the respective reactions. This is now Figure 1. Here, the reader gets a better idea of where ROS are produced. 

As requested by another reviewer, I also made some additions to the text. All changes are highlighted in yellow.

Thank you once more for your kind feedback.

Best regards,

Nana-Maria Grüning

Reviewer 4 Report

Comments and Suggestions for Authors

Tghids manuscript is a timely and well-written review of genetic disorders involved or leading to "oxidative stress". 

It however lacks some important actors both in the generation and in the scavenging of ROS.                                                                                            Regarding the generation of ROS, other enzymes such as xanthine oxidase and myeloperoxidase also contribute in a significant fashion to ROS production. In addition, the NO synthases NOS2 and NOS3 can also generate superoxide under certain conditions. For NOS2, this occurs when the cellular stores of Arg in macrophages are depleted and for NOS3, when it is uncoupled from BH4. Both of these are important pathophysiological mechanisms.                                                                                                               Regarding the "anti-oxidative" mechanisms, the Haber-Weiss reaction where superoxide interacts with Fe III, yielding Fe II and dioxigen i.e. the opposite of the Fenton reaction, should be mentioned.                                                             More important however is the "anti-oxidative" role of nitric oxide. Indeed, NO is the major substrate of superoxide ions as the latter reacts approximately 3 times faster than it is dismutased by SOD. Since its intracellular concentration also is 100- to 1000-fold higher than that of superoxide under physiologial concentrations, it is the primary cellular  "anti-oxidative" mechanism. No is primarily generated by the three NOSs. Polymorphisms with loss or gain of function have been reported for all three isoforms and mutations for NOS1 and NOS2. Interestingly, these variants and mutations have been linked to different diseases, most of which involve disturbed redox signalling.                                                                                       There are other interesting features of these NOSs: NOS2 can shift from NO to superoxide generation when intracellular arginine stores are depleted, which occurs rapidly when macrophages are activated and NOS3 will do the same when it becomes uncoupled from BH4, which occurs under conditions of "oxidative stress".

Redox signalling and anti-oxidative defense can thus obviously not be handled ignoring or not mentioning the primordial and essential role of NO. This review should therefor also address genetic disorders of NOSs as well as those of xanthine oxidase and myeloperoxidase.

Author Response

Dear Sir or Madam,

thank you very much for your kind feedback to our manuscript and your suggestions to improve the content of the text.

In the following I will address you feedback point by point.

"It however lacks some important actors both in the generation and in the scavenging of ROS. Regarding the generation of ROS, other enzymes such as xanthine oxidase and myeloperoxidase also contribute in a significant fashion to ROS production."

I included these enzymes in the sections "sources of ROS" and in the newly designed Figure 1. Additions to the text are highlighted in yellow.

"In addition, the NO synthases NOS2 and NOS3 can also generate superoxide under certain conditions. For NOS2, this occurs when the cellular stores of Arg in macrophages are depleted and for NOS3, when it is uncoupled from BH4. Both of these are important pathophysiological mechanisms."   

Thank you very much for pointing out these interesting features of NOS enzymes. I included the information in the text (highlighted as yellow) and in Figure 1.

"Regarding the "anti-oxidative" mechanisms, the Haber-Weiss reaction where superoxide interacts with Fe III, yielding Fe II and dioxigen i.e. the opposite of the Fenton reaction, should be mentioned."            

Both Fenton and Haber-Weiss reaction have now been mentioned in the text as well as in Figure 1.                                                

"More important however is the "anti-oxidative" role of nitric oxide. Indeed, NO is the major substrate of superoxide ions as the latter reacts approximately 3 times faster than it is dismutased by SOD. Since its intracellular concentration also is 100- to 1000-fold higher than that of superoxide under physiologial concentrations, it is the primary cellular  "anti-oxidative" mechanism."

The potential of NO to react with the superoxide anion is now mentioned in the section "the anti-oxidative machinery" as highlighted in yellow. 

"NO is primarily generated by the three NOSs. Polymorphisms with loss or gain of function have been reported for all three isoforms and mutations for NOS1 and NOS2. Interestingly, these variants and mutations have been linked to different diseases, most of which involve disturbed redox signalling."

Thank you very much for pointing to these genetic variants. As now added to the text, polymorphisms increase the risk for various diseases, clear monogenic disorders have however not (yet) been described for these genes. Given their important physiological role, they probably act as modifying factors for existing comorbidities. 

"There are other interesting features of these NOSs: NOS2 can shift from NO to superoxide generation when intracellular arginine stores are depleted, which occurs rapidly when macrophages are activated and NOS3 will do the same when it becomes uncoupled from BH4, which occurs under conditions of "oxidative stress". " 

This is an interesting aspect and an example of how certain enzymes can turn into ROS sources under certain conditions. Thank you for highlighting this aspect. I included this topic in the text and figure 1.

I hope we could address your feedback sufficiently.

Thank you once more for reviewing our manuscript.

Kind regards,

Nana-Maria Grüning                                                                               

Round 2

Reviewer 4 Report

Comments and Suggestions for Authors

Dear author,

Thank you for taking my comments and suggestions into account and completing your manuscript accordingly.

I feel that your review is now a very useful contribution to this field.

Kind regards